# Characterization of spatio-temporal dynamics of the constrained network of the filamentous fungus *Podospora anserina* using a geomatics-based approach

Clara Ledoux, Cécilia Bobée, Éva Cabet, Pascal David, Frédéric Filaine, Sabrina Hachimi, Christophe Lalanne, Gwenaël Ruprich-Robert, Éric Herbert[☯], Florence Chapeland-Leclerc[iD][☯]*

CNRS, UMR 8236 – LIED, Université Paris Cité, Paris, France

☯ These authors contributed equally to this work.
* florence.leclerc@u-paris.fr

**Data Availability Statement:** All relevant data are within the manuscript and its Supporting information files.

## Abstract

In their natural environment, fungi are subjected to a wide variety of environmental stresses which they must cope with by constantly adapting the architecture of their growing network. In this work, our objective was to finely characterize the thallus development of the filamentous fungus *Podospora anserina* subjected to different constraints that are simple to implement *in vitro* and that can be considered as relevant environmental stresses, such as a nutrient-poor environment or non-optimal temperatures. At the Petri dish scale, the observations showed that the fungal thallus is differentially affected (thallus diameter, mycelium aspect) according to the stresses but these observations remain qualitative. At the hyphal scale, we showed that the extraction of the usual quantities (*i.e.* apex, node, length) does not allow to distinguish the different thallus under stress, these quantities being globally affected by the application of a stress in comparison with a thallus having grown under optimal conditions. Thanks to an original geomatics-based approach based on the use of automatized Geographic Information System (GIS) tools, we were able to produce maps and metrics characterizing the growth dynamics of the networks and then to highlight some very different dynamics of network densification according to the applied stresses. The fungal thallus is then considered as a map and we are no longer interested in the quantity of material (hyphae) produced but in the empty spaces between the hyphae, the intra-thallus surfaces. This study contributes to a better understanding of how filamentous fungi adapt the growth and densification of their network to potentially adverse environmental changes.

## Introduction

Fungi are of crucial importance for plant and microbial nutrition with key roles in decomposition and nutrient recycling. Most fungi are able to use environmental resources by the production of extracellular enzymes for the decomposition of the organic matter, even in presence of

**Funding:** Clara Ledoux is supported by a PhD scholarship from the Doctoral School MTCI (ED 563). This work is supported by the IdEx Université Paris Cité (ANR-18-IDEX-0001) and by ANR21-CE40010-01. The funders had no role in study design, data collection and analysis, decision to publish, or preparation of the manuscript.

**Competing interests:** No competing interest.

other organisms and in stressful environments [1]. The colonization success of saprophytic fungi thus relies on their ability to establish and then persist within the resource until reproduction and dissemination [2]. The achievement of filamentous fungi in colonizing such terrestrial ecosystems can be largely attributed to their flexible morphology, and more specifically to their ability to form an interconnected hyphal network, the mycelium, based upon some fundamental cellular processes, such as hyphal tip growth, septation, hyphal orientation, branching and fusion (also known as anastomosis) [3]. Hyphal branching has also been well described and appears to both increase the surface area of the colony, which enhances nutrient assimilation, and to mediate hyphal fusion events that are important for exchange of nutrients and signals within the same colony [4]. Therefore, the architecture of the fungal network is clearly not fixed, but must continually adapt to local nutritional or environmental cues, damage or fungivore attacks.

In this context, for many years, various studies have been carried out to better characterize the growth dynamics and topology of the fungal network and the impact of different environmental constraints on its growth. The main challenges in this type of approach are i) to find *in vitro* conditions that are not too far from the usual lifestyle of the fungus, but simple enough to collect robust and interpretable data and ii) to develop fully automated systems allowing rapid and efficient morphological quantification of fungal hyphae dynamics through image analysis [5]. In [6], imaging approaches and software tools have been developed to quantitatively characterize nutrient transport and network formation in foraging mycelia of the basidiomycete *Phanerochaete velutina* over a range of spatial scales. In particular, on a millimetre scale, thanks to a combination of time-lapse confocal imaging and fluorescence recovery after photobleaching, this approach allowed the authors of [6] to quantify the rate of diffusive transport through the vacuole system in individual hyphae. More recently, experimental setups based upon an automated image analysis allowed for a functional and quantitative description of fungal growth dynamics, through the extraction of the most studied fungal characteristics such as the total length of the mycelium, the number of nodes and apexes and the area of the mycelium [7–9]. A similar approach was described in [10] which proposed an automated, continuous video microscopy tracking of hyphal growth that affords quantitative analysis of growth rate and morphology in the ascomycete *Trichoderma atroviride*. In different works, image analysis has been used to study the impact of environmental conditions, such as temperature, relative humidity, aeration, synthetic substrate composition, and/or nutrient concentration, on the growth of filamentous fungi at different stages, starting from germinating spores or mycelia fragment. For example, automated image analysis was used to assess the impact of environmental conditions, *i.e.* four temperatures and four relative humidities, on the growth dynamics of two economically important fungal species, *Coniophora puteana* and *Rhizoctonia solani* [11]. In a similar way, the interactive effects of a 4°C temperature increase and soil invertebrate grazing (as collembola) on multispecies interactions between cord-forming basidiomycete fungi *Hypholoma fasciculare*, *P. velutina* and *Resinicium bicolor* emerging from colonized beech (*Fagus sylvatica*) wood blocks in soil microcosms has been studied by digital image analysis approaches [12]. Probably one of the most exciting developments in this area in recent years are the microfluidic approaches, usually combined with time-lapse fluorescence microscopy to visualize the dynamic response of the mycelium to various constraints. In particular, some very nice works explored in this way fungal cell growth in confined environments with geometrical characteristics more similar to native microbial habitats than traditional observation in culture plates. Then, the authors of [13] monitored hyphal growth behavior and strategies of various basidiomycete litter decomposing species in a microfabricated "Soil Chip" system that simulates principal aspects of the soil pore space and its micro-spatial heterogeneity. Such an approach has been also used in [14] which combined a tailor-made microfluidic

platform with time-lapse fluorescence microscopy to visualize the dynamic response of the mycelium of the basidiomycete *Coprinopsis cinerea* to two different stimuli, *i.e.* access to nutrients and attack by fungivores.

*Podospora anserina* is a coprophilous filamentous ascomycete, a large group of saprotrophic fungi, that mostly grows on herbivorous animal dungs and plays an essential role within this complex biotope in decomposing and recycling nutrients from animal feces [15]. *In vivo*, *P. anserina* evolves in a highly competitive habitat where several dozens of species are present and feed on partially degraded plant material. So, growth of *P. anserina* in this confined microcosm appears to be highly challenged by resource-limited and patchy dung environment [16]. Moreover, *P. anserina* has long been used as an efficient laboratory model to study various biological phenomena, especially because it grows rapidly at a rate of 7 mm/ day on standard medium, it accomplishes its complete life cycle in only one week, leading to the production of ascospores, and it is easily usable in molecular genetics, cellular biology and cytology [16]. In a previous work [17], we have developed an automated and reproducible experimental device to track the hyphal network construction of *P. anserina*. Such a system allowed us to monitor temporal series of pictures of the fungal thallus, followed by a robust image analysis process. This led to the extraction of a set of reproducible quantitative parameters, such as the total length of the mycelium, the number of nodes and apexes, as well as the linear densities and the intra-thallus area distribution. Such a systematic spatial and temporal exploration enabled us to estimate a set of key physiological features of the fungal network, such as the anastomosis rate, the branching dynamics and the spatio-temporal patterning of the network grown in standard conditions. More recently, we proposed an efficient method to quantitatively analyze the angular distribution of branching among the fungal network [18].

In this work, based on the experimental system described in [17], we propose to deepen the characterization of the fungal network by applying different abiotic constraints on the growing fungal network. Then, several constraints have been tested in this study and compared to optimal standard conditions for the growth of *P. anserina in vitro*: i) *low-nutrient medium*, as it has been shown that in nature, fungi commonly grow in conditions of nutrient depletion, but are able to optimally use the scare resources [19]; ii) *high-osmotic medium* through the use of potassium chloride (KCl), which changed the water balance between the cell and the extracellular environment, and caused a sudden loss in the turgor pressure [20]; iii)*intense lighting of visible light*, as light sensing with photoreceptors is a source of information for fungi to adapt to environmental conditions [21, 22]; and iv) *low* and v) *high temperatures*, as fungi are constantly subject to temperature fluctuations in their natural habitats [20].

Beyond the extraction of quantitative quantities (apex, node, length) commonly used in such experiments, we have developed an original geomatic approach, using Geographic Information System (GIS). GIS are computer system tools used in the geomatic discipline, and involve a wide range of methods for collecting, managing and analyzing data gathered by various devices (including satellites, drones, meteorological stations, governmental census. . .). The connectivity and arrangement of a network is known as its topology in GIS. Thus, road transport networks, for example, have various specific topologies denoting their structures in terms of edges, vertices, paths, and cycles [23]. Analogies between the fungal network and some geographic networks may be drawn, and GIS may be easily transposed to the scale of the Petri dish for analyzing the thallus growth and densification processes. Thanks to an interdisciplinary work within the scope of this contribution, with geographers, biologists and physicists, we have initiated a geomatic approach, unusual in the field of microbiology, to characterize the fungal network of *P. anserina* in a previous work [17]. Here, we have shown that this original approach constitutes a very powerful tool for

characterizing various behaviors in terms of densification of the fungal network depending on the stress applied, in an efficient and quantitative manner. This could be a very useful analysis tool to complement more classical mechanistic approaches at the cellular level, contributing to a better understanding of the mechanisms underlying fungal stress adaptation and regulation of these responses.

## Materials and methods

### Strain, media, culture conditions and preparation of biological samples

The *P. anserina* strain used in this study is the "S" wild-type strain for all experiments [24]. All the protocols including standard culture conditions and medium composition for this microorganism can be accessed at http://podospora.i2bc.paris-saclay.fr (see also [16]). Ascospores expelled from mature perithecia were collected and placed on a germination medium during few hours to initiate the ascospore germination and recovered with a cellophane sheet to guide the growth in two dimensions (2D), as described in [17]. The germinated ascospore constituted the initial point of the growing mycelium that we then observed. In standard culture conditions, the cellophane sheet containing the germinated ascospore was transferred to a Petri dish containing the standard M2 medium, as used in [17]. According to experiments, the M2 medium could be replaced by either the M0 medium in which the dextrin carbon source was removed or by the M2 medium supplemented with 0.3 M of potassium chloride (KCl). To observe the thallus at both the macroscopic and microscopic scales, the Petri dish was then placed in an insulated enclosure where the temperature, hygrometry and light intensity are controlled. Under standard culture conditions, the temperature was regulated at 27˚C (optimal temperature of *P. anserina* growth), as described in [17]. The lighting of the sample was obtained by transmission through the culture medium, using a collimated white LED MCWHL6-C1 (Thorlabs) combined with an absorptive neutral density filter (Thorlabs, NE2R30B, optical density of 3). The parameters of the lighting were 548 mW, 600 mA and 67 lux. This intensity of visible light is comparable to that observed on a dark, overcast day, see [25]. No effect was found when comparing the growth dynamics of the parameters (number of apexes, total length of the mycelium) obtained in the dark (not shown) with those obtained with this lighting, which is therefore considered neutral. Lastly, the hygrometry was regulated around 60% in the enclosure to prevent desiccation of the solid culture medium in the Petri dish [16]. According to experiments, the temperature could be adjusted to 35˚C (8˚C above the optimal temperature) or 19˚C (8˚C under the optimal temperature) to test the effect of temperature variations on the growth and organization of the fungal thallus. Similarly, in order to test the effect of intense lighting on the fungal network, the absorptive neutral density filter was removed, so that the light intensity was measured to $66.10^3$ lux. This intensity of visible light is comparable to that observed on a summer day with the sun at its highest. It is 1000 times brighter than the neutral lighting described for the standard conditions [25].

### Macroscopic observations

The thallus diameters were measured after 2, 4 and 7 days of growth, except with experiments with intensive lighting. In this last case, diameters were measured after 2 and 4 days of growth. Beyond this period, the surface of the thallus exceeded the surface of the light beam. In addition, pictures of the whole thallus were taken with the device UVIDOC HD5–20MX Electronic Darkroom (Uvitec) after 4 days of growth. For each experiment, three independent replicates were carried out.

## Experimental setup for microscopic observations

We used the experimental device described in [17] to observe the hyphal network of *P. anserina* growing on a Petri dish for approximately 20 hours. The setup allowed us to obtain an image of the complete growing network (called a "panorama" in what follows) by moving the sample along two axes thanks to precision micro-control plates. The $3 \times 4$ individual images (called "tiles") were assembled to reconstruct a panorama. Tiles ($3.99 \times 3.34$ mm) were captured using a camera combined with a telecentric objective (2× magnification, 65 mm working distance). The distance by which the sample was moved between two successive tiles allowed for an overlap by 20%, so that a complete panorama covered an area of $11.7 \times 11.4$ mm$^2$. The image resolution is 1.63 μm/px. For each experiment, three independent replicates were carried out.

## Image processing for extraction of the vertices

The classical image processing was described in [17]. The tiles were assembled to reconstruct a panorama, *i.e.* a greyscale image of the complete hyphal network at a given time. The successive panoramas were then binarized and vectorized to extract a collection of vertices, whose number of connections to other vertices allowed us to determine their nature: apex, hypha or node. Thus, for each experiment (a collection of successive panoramas), it was possible to extract the time evolution of the number of apexes $A(t)$, the number of nodes $N(t)$ and the total length of the mycelium $L(t)$ over time For this, we used the following expression to fit the evolution of each quantity, as described in [18]:

$$X(t) = X^0 \, 2^{(t+t_0)/\tau} \tag{1}$$

where $X$ stands respectively for $A$, $N$ or $L$. The characteristic growth time is $\tau$ and $t_0$ is the temporal offset corresponding to the transition between the end of the lag phase and the first observation. For the fitting range, we excluded the data at times lower than about 1 hour as well as the last experimental points that deviate from the exponential phase. $X^0$ are the respective parameters $A^0$, $N^0$ and $L^0$, corresponding to the growth onset given the temporal offset. As explained in [18], the value of $t_0$ was manually adjusted in order to obtain simultaneously $A^0$, $N^0$ and $L^0$ in the range of the respective expected values (respectively 3, 1 and in the range 10 to 20 hypha diameters).

This work has been carried out for the six experimental conditions (one standard condition and five abiotic constraint conditions, see Table 1), each of which comprises a set of three independent experiments (called triplicate in the following). All the values extracted from the fit

**Table 1. Conditions tested on the fungal thallus.**

| Conditions | Nature of stress | Culture medium | Lighting | Temperature |
|---|---|---|---|---|
| condition 0 | No | M2 | neutral | 27˚C |
| condition 1 | nutrient | M0 | neutral | 27˚C |
| condition 2 | osmotic | M2 + KCl (0.3 M) | neutral | 27˚C |
| condition 3 | lighting | M2 | intense | 27˚C |
| condition 4 | low temperature | M2 | neutral | 19˚C |
| condition 5 | high temperature | M2 | neutral | 35˚C |

For standard conditions, see [17]. Condition 0 corresponds to the standard growing conditions. The light intensities for neutral and intense lighting was measured to 67 lux and 66.10$^3$ lux, respectively. Details in Section Materials and methods.

**Table 2. Offsets and mean growth rate exponents for *A*, *N* and *L*.**

| Condition | $t_0$ [h] | $\tau_A$ [h] | $\tau_N$ [h] | $\tau_L$ [h] |
|:---:|:---:|:---:|:---:|:---:|
| (0) | 0.31 | 2.09 ± 0.06 | 1.62 ± 0.04 | 1.96 ± 0.07 |
| (1) | 0.51 | 2.91 ± 0.09 | 2.36 ± 0.07 | 2.79 ± 0.11 |
| (2) | 0.58 | 3.01 ± 0.09 | 2.30 ± 0.06 | 2.72 ± 0.10 |
| (3) | 2.67 | 3.33 ± 0.18 | 2.30 ± 0.08 | 3.03 ± 0.23 |
| (4) | 0.39 | 3.66 ± 0.16 | 2.57 ± 0.08 | 3.21 ± 0.17 |
| (5) | 1.76 | 2.00 ± 0.06 | 1.62 ± 0.04 | 2.04 ± 0.09 |

Summary for each conditions (average of the triplicates) offset values $t_0$ and the mean growth rate exponents (with uncertainties to one standard deviation) for the number of apexes *A*, the number of nodes *N* and the total length *L* extracted from data in each condition 0 to 5. The $\chi^2$ values were all found to fall into the range 1–10. The uncertainty on $t_{0,i}$ was estimated to be about 1 h. More details can be found in S1 Table.

are summarized in Table 2. Within a condition, the three values of $X^0$ and $\tau$ are respectively compatible with each other.

## Geomatics

A geomatics-based approach was used to produce maps and metrics characterizing the growth dynamics of the networks, through the use of automatized GIS tools. The software used was ArcGIS Pro (version 3.0.3) and automation of treatments was carried out with the model builder interface. In this section, the terms below in square brackets correspond to the main GIS tools used. One set of panoramas per experimental condition has been chosen.

**Binarization and vectorization of the thallus.** The first GIS processing step, developed in [17], aimed at extracting thallus as a thin two-dimensional feature (polygon type) from grey-scale panoramas as follows: (i) the generation of a binary image using a threshold operation from the grey-scale panoramas [Raster Calculator]; at this stage, both the thallus and sparse speckles were defined by non-zero pixel values (Fig 1A); (ii) the conversion of the resulting binary image into a set of polygon feature classes: each group of contiguous cells with the same values (0 or 1) was converted into one polygon [Raster to Polygon function]; all polygons were stored in the same shapefile; (iii) the thallus polygon extraction was processed by selecting the polygon object that intersects the center of the mycelium (a predefined object with a point-type geometry that we placed at the center of the thallus) [Select Layer by Location].

**Extraction of intra-thallus areas.** The collection of intra-thallus areas $S_i$ were defined as all zero value polygons that were completely inside the thallus. These surfaces were computed for each panorama [select layer by location and select layer by attribute tools] and statistics extracted [Add Geometry Attributes tool], which are the sum, the mean and the number of $S_i$. The total surface *S* was defined as the sum of $S_i$. Note that this method excludes the outermost ring of the fungal thallus (areas not completely closed by hyphae) in the calculation of the total surface. Maps of $S_i$ were also produced (Fig 1B), using in this work a "natural breaks" discretization method (also called "Jenks optimization"). This classification method tends to reduce the variance within each class and maximize the variance between classes [26]. For each condition tested, one experiment from the triplicate has been processed. This approach allows for the observation of spatial heterogeneities in intra-thallus areas over a longer period of time than the previously described approach for the quantities *A*, *N* and *L* (see Fig 6 compared to Fig 4). This was made possible thanks to a different image processing procedure.

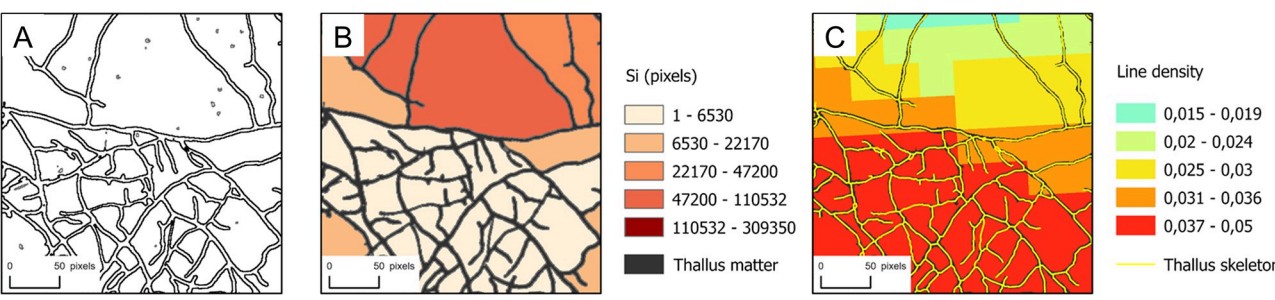

**Fig 1. GIS processing in a zoomed region showing different stages of treatments.** (A) Vectorization stage showing thallus polygon features. (B) Intrathallus areas shapefile and thallus matter. (C) Thallus skeleton polyline feature and line density map (using a 50-px resolution and a search radius of 200 px).

**Skeletonization of the thallus.** Linear features of thin rasterized thallus were computed from "clean binary images" in order to provide line density maps thereafter. The GIS Thinning tool [Thin] algorithm used is detailed in [27]. The method consisted in reducing the width of the fungal hyphae (the input image being the "clean binary map") to a set of non-zero contiguous values of 1 px wide. For this processing, the maximum thickness of input linear features had to be specified and, according to what we observed in the binary images, the parameter was fixed at five times the cell size (5 px). Finally, the resulting images of thallus skeletons were vectorized into polyline features [Raster to polyline conversion tool].

**Maps of thallus line density.** We also processed for each experiment rasters of thallus polylines density [Line Density spatial analyst tool]. To do this, we had to define an output raster cell size and a search radius, which were respectively fixed at 50 px and 200 px (Fig 1C). For every cell, the search radius center was placed on the center of the cell. The methodology is the following [28]: for each raster cell of the output image, it consists in adding lengths of hypha segments that falls within the search radius, then to divide the total length by the circle area. The line density pixel values were further grouped into 5 classes using a natural break discretization method, thus allowing us to better observe the spatial variation of linear densities. Maps provide two types of information: i) observation of spatio-temporal densification processes, ii) the global morphology of thallus envelops (*i.e.* its size).

## Uncertainties

We assume that the number of apexes and nodes ($A$ and $N$) are random variables following a discrete Poisson distribution and we associate the standard deviation corresponding to this distribution with their uncertainty. The total length $L$ is about $A + N$ segments of average length $\langle \ell \rangle$. An estimation of the uncertainty associated with the measure of the total length $L$ was derived as $\sigma_L = \sqrt{(A + N)} \langle \ell \rangle$. We assume that the uncertainty associated to the acquisition time $t$ is $\delta t = 1$ min, corresponding to the time needed to acquire all the tiles of a panorama.

For the total surface of the mycelium $S$, the uncertainties were calculated by bootstrapping. The uncertainties associated to the counting of the number of surfaces $S_i$ were estimated independently for each time step. We considered each count as a Poisson random variable. Recall that the standard deviation of a Poisson random variable with mean λ is $\sigma = \sqrt{\lambda}$. We assumed the uncertainties associated to the densities of line to be Poisson as well.

## Results

### Macroscopic observations

The different stresses applied to the fungal thallus are listed in Table 1. Condition 0 corresponds to standard culture conditions classically used for optimal growth and reproduction of *P. anserina* (M2 culture medium, 27˚C, standard lighting [16]). It should be noted that stresses were applied to the fungal thallus throughout the experiment and that they were all abiotic but varied in nature. Thus, we tested constraints of nutrient (low-nutrient medium, condition 1), osmotic (medium with KCl, condition 2), luminosity (intense lighting, condition 3) and temperature (19˚C and 35˚C, conditions 4 and 5, respectively). This allowed us to study how the thallus adapt to very different stresses.

First, we performed a macroscopic observation of the growth of the fungal thallus under different conditions and during 7 days. The experimental conditions are strictly identical to those used for the microscopic characterization of the fungal network. Pictures of the whole fungal thalli in Petri dish were taken after 4 days of growth under different conditions (Fig 2A). The evolution of fungal thallus growth was performed by measuring the diameters of mycelia after 2, 4 and 7 days of growth (Fig 2B). As expected, condition 0 is the most favorable for growth and will be considered as the control (most extensive thallus, fairly dense, regular, not pigmented mycelium). All other conditions limit the extent of mycelia on the medium and seem to induce growth stress. The thallus under intense ligthing (condition 3) is the most affected, with a mycelium of very reduced diameter, of extremely dense, thick and pigmented appearance. The low-nutrient medium (condition 1), osmotic stress (condition 2), and low temperature (condition 4) all result in a more tenuous mycelium than the control. It is interesting to note the difference in appearance between the thallus submitted to a lower (condition 4) or higher (condition 5) temperature of 8˚C compared to the optimal temperature. The thallus that grew at low temperature is less dense than at high temperature, the latter showing an irregular growth front and a smaller diameter. The observations made on Petri dishes show that the fungal thallus is macroscopically different according to the constraints applied but these observations remain qualitative and do not inform us about the effect on the organization and the growth dynamics of the hyphal network.

### Apexes, nodes and length growth dynamics

For each condition tested, three sets of panoramas were obtained over 20 h of growth under controlled conditions. From these panoramas, we extracted the number of apexes $A$, the number of nodes $N$ and the total length of the mycelium $L$ over time. An example of a panorama obtained after 14 h of growth under standard conditions (condition 0) is presented in Fig 3A. The identification of apexes, nodes and hyphal segments on a small region of mycelium is also shown Fig 3B. Note that the nodes can be real biological nodes, *i.e.* from a branching, but can also correspond to overlaps between two hyphae.

The evolution of these three quantities (number of apexes $A$, number of nodes $N$ and total length of the mycelium $L$) allowed us to characterize the growth dynamics of each network.

The mean temporal offsets $t_0$ and the mean growth rate exponents $\tau$ are summarized in Table 2 for each condition. The curves obtained with the mean parameters of the fit on the considered data range are shown in Fig 4. Note that the extraction of quantities is no longer possible when the thallus is too dense, which explains why the different curves do not all stop at the same time. Moreover, as the mycelium behaves a branching network in the form of a binary tree [17, 18], we would expect to have a similar number of apexes $A$ and nodes $N$, leading to $\tau_A \backsim \tau_N$. However, $\tau_N$ is found systematically lower than $\tau_A$, because $N$ includes

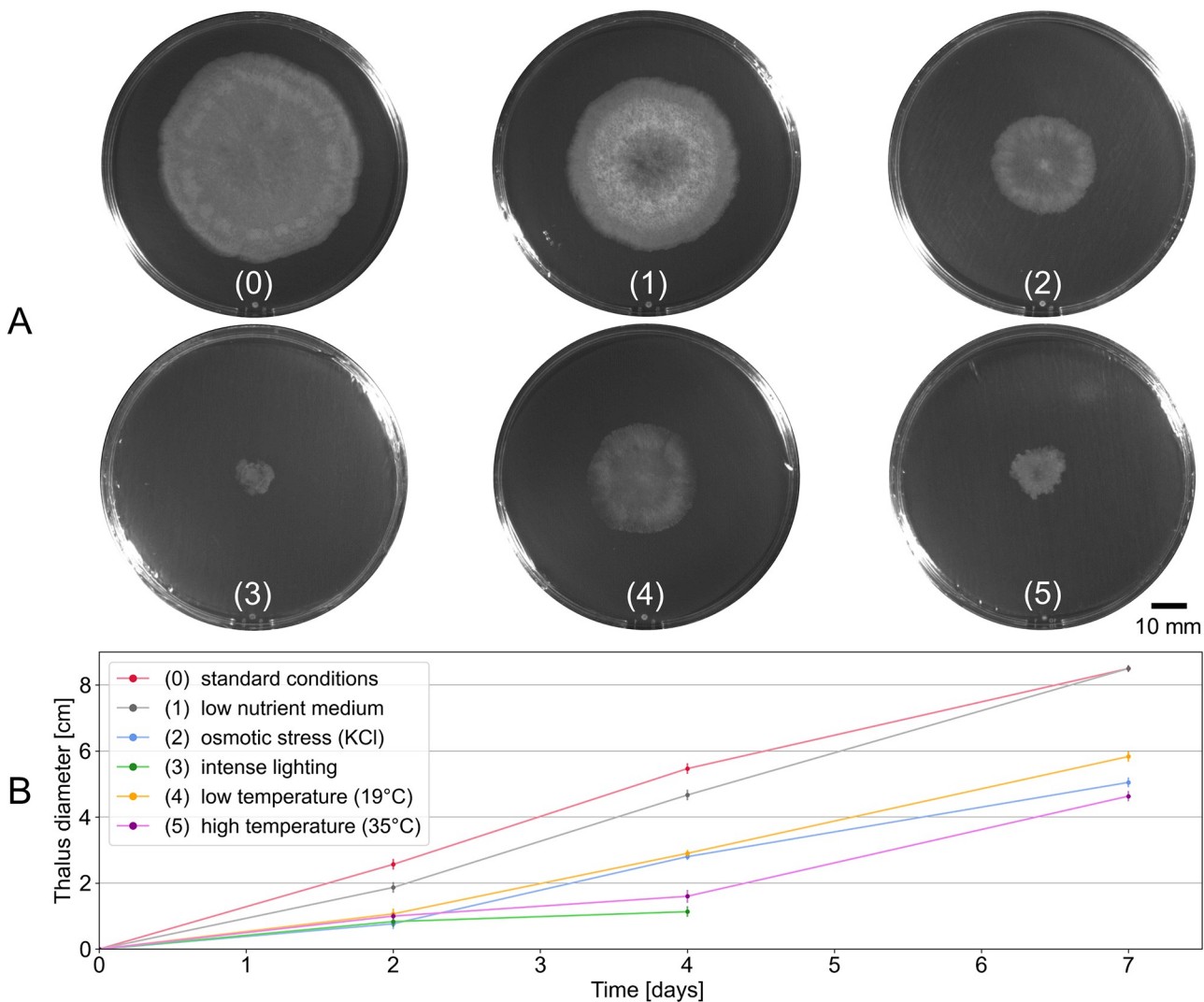

**Fig 2. Observation of the thallus in different growth conditions 0 to 5 at the macroscopic scale.** (A) Thallus on Petri dishes in different conditions of growth, 4 days after the germination of the ascospore on a membrane of cellophane (scale bar: 10 mm). (B) Thallus diameters over time of growth, with the errorbars corresponding to the standard deviations calculated from the three experiments. (0) condition 0 (standard conditions, see [17]); (1) low-nutrient medium; (2) osmotic stress (KCl); (3) intense lighting (diameter measured for days 2 and 4 only, see Section Materials and methods); (4) low temperature (19˚C); (5) high temperature (35˚C). Experimental points were connected by straight lines.

branching points, as well as anastomosis and intersections, *i.e.* overlaps between hyphae, making $N$ values higher than $A$. Also note that the uncertainty on the number of nodes $N$ is therefore underestimated.

The temporal offset values $t_0$ show that the constraints applied on the thallus for conditions 3 and 5 induce a delay before the exponential growth phase compared to standard culture conditions (condition 0). We also observe different growth dynamics between all conditions, quantifiable with the growth rate exponents (Fig 4 and Table 2). Compared to condition 0, the growth dynamics of $A(t)$, $N(t)$ or $L(t)$ are always affected except in the case of high temperature (condition 5). For all the other conditions, the stress causes a reduction in growth activity. Fungal thallus under high lighting (condition 3) or low temperature (condition 4) are the most affected with $A$, $N$ and $L$ values remaining the lowest over time, but are not distinguishable

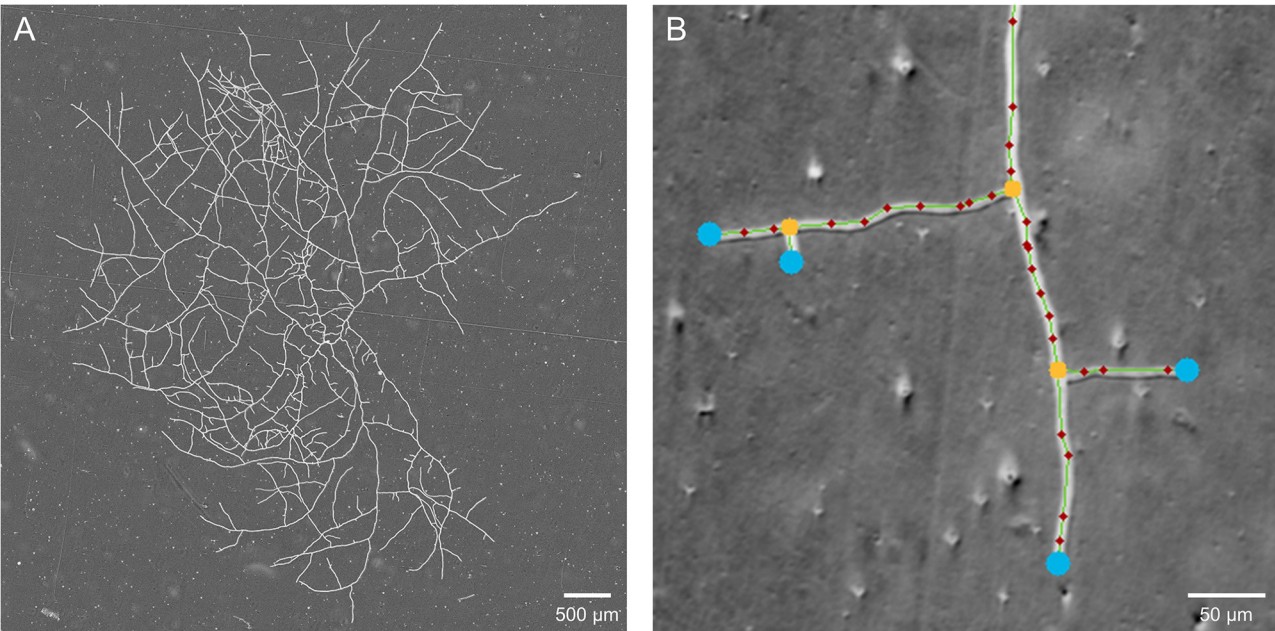

**Fig 3. Panorama of the thallus and image processing.** (A) Panorama of the mycelium of *P. anserina* 14 h past the ascospore germination in standard conditions (scale bar: 500 μm). (B) Hyphae superimposed with the output of the vectorization process. Blue dots are apexes and yellow dots are nodes connecting three branches. The connected paths of red dots constitute the hyphae (scale bar: 50 μm).

from each other taking into account the uncertainties. Fungal thalli under nutrient stress (low-nutrient medium, condition 1) or osmotic stress (condition 2) have slightly lower doubling times than under conditions 3 and 4 for all three quantities $A$, $N$, $L$ (see Table 2 and S1 Table for more details).

## Geomatics-based approach

**Intra-thallus areas.** As a first step, we mapped the fungal networks after 15 h of growth for the six conditions. We show in Fig 5 the intra-thallus surfaces colored according to the

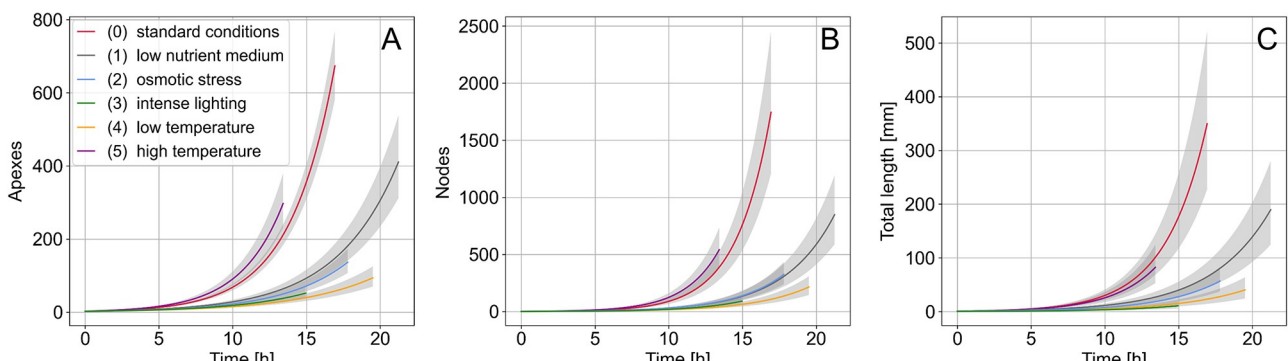

**Fig 4. Number of apexes (A), number of nodes (B) and total hyphal length (C) as a function of time in conditions 0 to 5.** Solid lines represent the best mean fit parameters for three replicates in each condition (see Table 2). The grey shadowing thickness quantifies the associated uncertainties to one standard deviation.

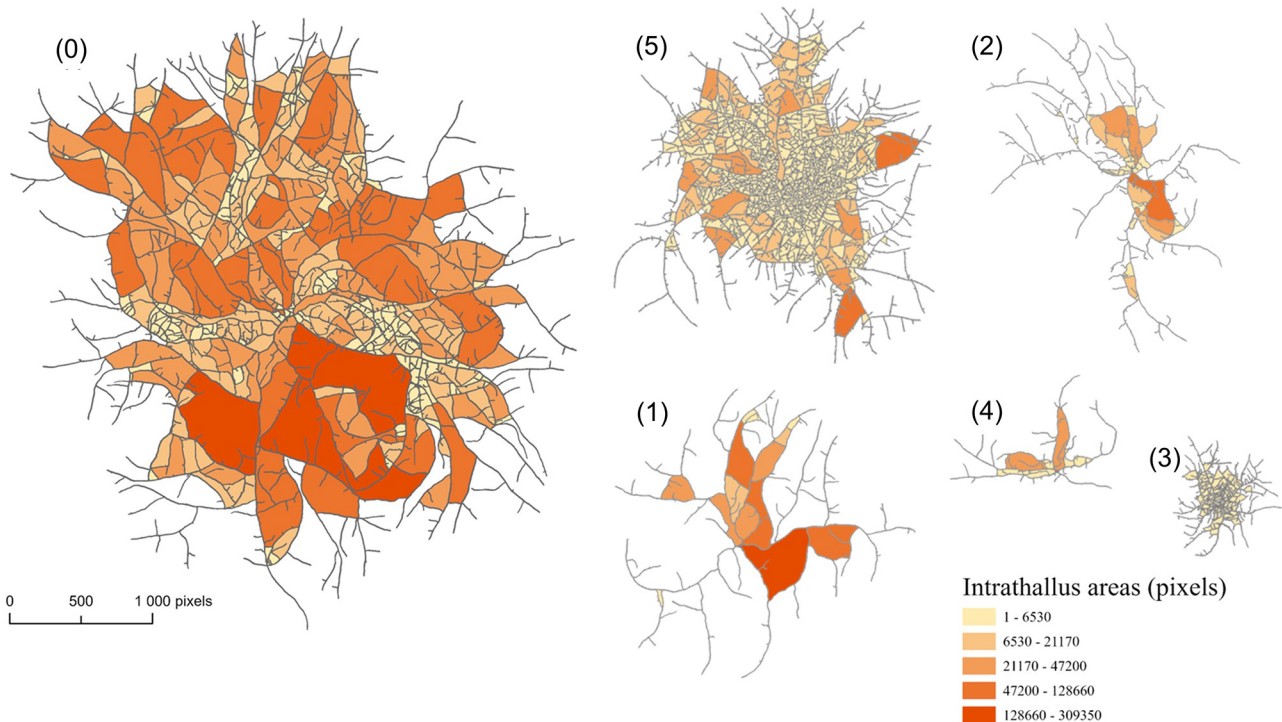

**Fig 5. Colored intra-thallus areas on mycelia in different conditions of growth.** (0): standard conditions; (1): low-nutrient medium; (2): osmotic stress; (3): intense lighting; (4): low temperature; (5): high temperature. Panoramas shown here correspond to the same growth time of 15 h taking into account the temporal offset $t_{0,i}$. Areas have been discretized into five classes.

surface size. As expected, condition 0 seems to be the most favorable to growth, with a well-developed thallus. The collection of $S_i$ surfaces presents a wide range of sizes. We also distinguish three main axes of densification, that aggregate a large number of small areas. This probably corresponds to the first three hyphae, as described in [17]. The largest surfaces are located favourably towards the outer ring of the thallus. All stressed conditions show fungal thalli much smaller in diameter (especially in conditions 3 and 4), less circular (especially in conditions 1, 2 and 4) and with very small intra-thallus surfaces (especially in conditions 3 and 5). Also note that because the total surface area $S$ is built here from the surfaces formed between the hyphae, the network density impacts the $S$ value. For example, $S$ is much lower in condition 1 compared to condition 5 (colored surface), whereas we would obtain equivalent quantities by relying on the convex hulls, *i.e.* the smallest convex shape enclosing all objects.

To extend the analysis, the temporal evolution of the total intra-thallus surface $S$ for each condition 0 to 5 was performed (Fig 6A). As hyphae grow radially from the ascospore into the available space around it, the surface area covered by hyphae necessarily increases over time, but more or less rapidly depending on the growth condition. For example, after 21 h of growth, the fungal network in condition 0 (control) covers 40 mm$^2$, against 0.6 mm$^2$ for condition 3, and 10 mm$^2$ for condition 1. A linear fit was performed on each set of experimental points to characterize the surface growth depending on the condition. The fitting range starts at a user-selected time $t_{min}$ (different for each condition) and extends to the last experimental point $t_{max}$ (see Table 3). This range was chosen to exclude initial points clearly outside the linear phase. In Table 3 are summarized the values of the linear coefficients $a$ of the fit lines. We made use of R squared $R^2 = 1 - \frac{SS_r}{SS_t}$, with $SS_r$ the residual sum of squares and $SS_t$ the total sum of squares

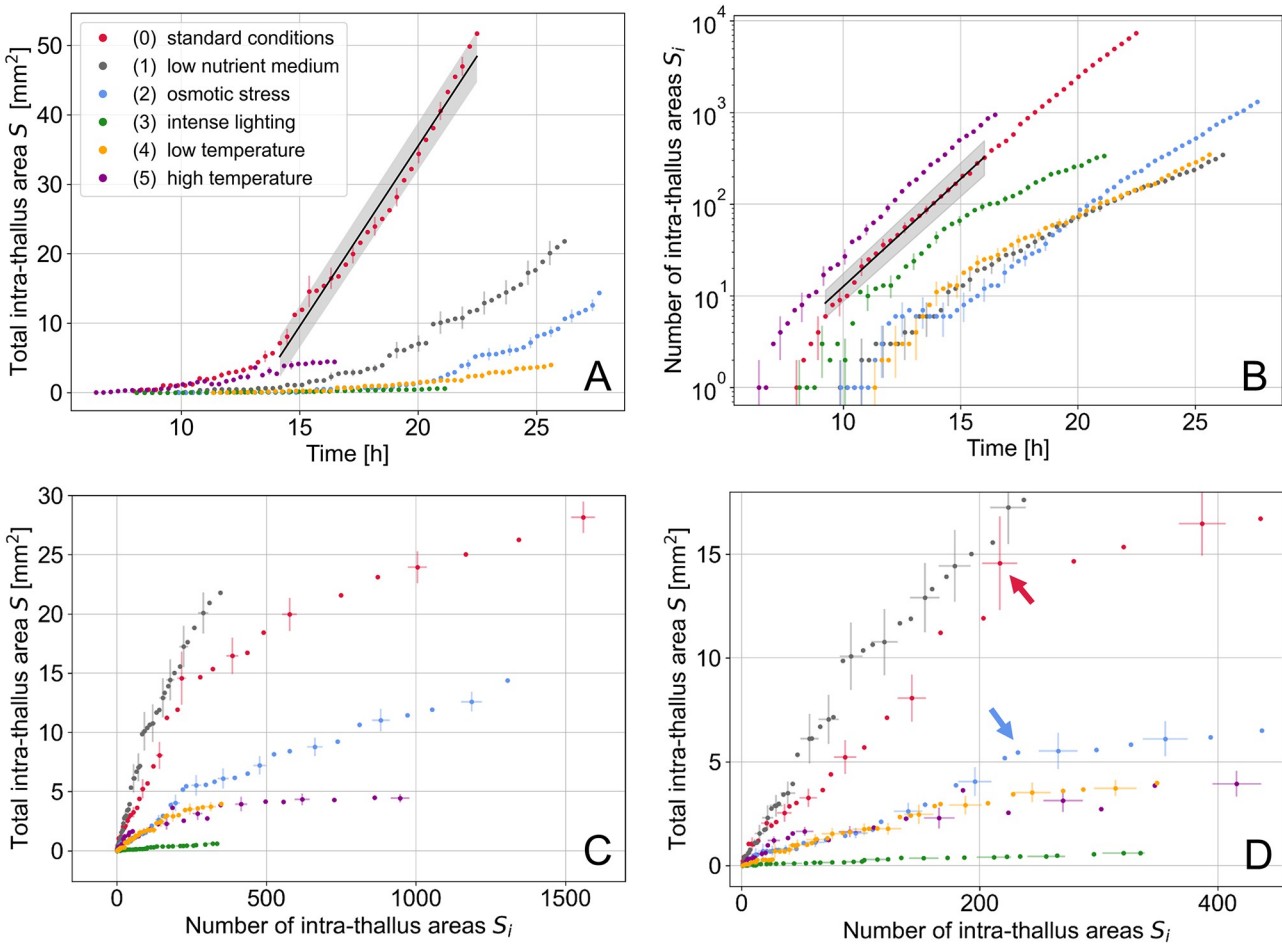

**Fig 6. Dynamical evolution of intra-thallus areas for the six conditions tested.** (A) Time evolution of the total intra-thallus area $S$ for experiments in each of the six conditions. The data points are represented with their errors (one point out of three) (see Section Materials and methods). The solid black line represents the linear fit and the grey shadowing quantifies one standard deviation. The fit was represented for condition 0 only to keep the figure clear. All the fitting values are in Table 3. (B) Time evolution of the number of intra-thallus areas $S_i$ (semi-logarithmic scale). See Table 4 for fitting values. (C) Evolution of the total intra-thallus area $S$ as a function of the number of surfaces $S_i$. The figure has been enlarged around the area of interest. Condition 0 only is cropped, with the maximum abscissa and ordinate point at 7330 and 52 respectively. (D) Same as (C) with an enlargement on the first points. Both arrows indicate the breaking points of the slopes visible in conditions 0 (red arrow) and 2 (blue arrow) respectively.

**Table 3. Linear fit of the temporal evolution of S.**

| Condition | $t_{min}$ [h] | $t_{max}$ [h] | $a$ [mm$^2$.h$^{-1}$] |
|:---:|:---:|:---:|:---:|
| (0) | 14 | 22.5 | 5.19 ± 0.09 |
| (1) | 17 | 26.2 | 2.04 ± 0.07 |
| (2) | 20 | 27.6 | 1.52 ± 0.06 |
| (3) | 13 | 21.1 | 0.06 ± 0.01 |
| (4) | 16 | 25.6 | 0.36 ± 0.03 |
| (5) | 8 | 16.5 | 0.53 ± 0.04 |

The temporal evolution of $S$ was fitted with a linear function on a range of values between $t_{min}$ and $t_{max}$ for each condition 0 to 5. Both $t_{min}$ and $t_{max}$ are dependent on $t_{0,i}$. $a$ is the slope of the linear fit with its uncertainties (one standard deviation). The values of the coefficient of determination $R^2$ (see text) are in the range 0.94–0.99.

to discuss the quality of the fit. As expected, the fungal network under standard conditions (condition 0) grows more than twice as fast as the network under nutrient-poor conditions (condition 1). The other growth conditions affect the surface extension of the thallus even more. In addition, we found that the lag phase, for which the increase in surface $S$ is very small (before moving to the linear phase), is the shortest for the control (condition 0) and high temperature (condition 5) experiments.

The Fig 6B represents in semi-logarithmic scale the number of intra-thallus surfaces $S_i$ extracted over time for each condition. We observe that the growth of the mycelium induces an increase in the quantity of material (hyphae produced), but also a complexification of the network. This complexity is reflected in the creation of new intra-thallus surfaces following overlaps or fusions events that are increasingly frequent over time. The evolution of the number of surfaces $S_i$ is fitted using an exponential function $\beta \exp(\alpha t)$. All fits are performed with $S_i$ in the range 5 to 336, this last value corresponding to the minimal number of surfaces reached among all the conditions at the end of the experiment, $i.e. S_i = 336$ (imposed by condition 3). The growth rate exponent $\alpha$ allows us to quantify the production of the surfaces $S_i$ over time (see Table 4), which is similar for conditions 0 and 5, with an earlier start of the exponential phase for condition 5. For the other conditions 1, 2, 3 and 4, we notice that surfaces formation is not simply delayed compared to the control (condition 0), but follows a different evolution, for which $S_i$ production is slower. We also observe that condition 4 (low temperature) induces a very different evolution of $S_i$ production than condition 5 (high temperature). Finally, the exponential exponents are closed for conditions 1 and 3. We have shown in Fig 6A that the total area $S$ is much smaller in condition 3 than in condition 1, meaning that the fungal network developed in condition 3 is very dense compared to condition 1 (see Fig 5).

Fig 6C shows that for a given total surface area $S$, fungal networks from conditions 2, 3, 4 and 5 are denser than the control (condition 0), as they have a greater number of intra-thallus surfaces $S_i$. On the other hand, condition 1 (low-nutrient environment) induces a network that is always less dense than the control (for a given total area). This graph has been enlarged on the first points (Fig 6D). For both conditions 0 and 2, an initial linear phase can be discerned up to about $S_i = 200$ (arrows), followed by a break in the slopes. During this linear phase, the total surface $S$ increases by adding new surfaces $S_i$ of equivalent size, $i.e.$ these are not divided by hyphae. The breakpoints indicate a new phase where the increase of $S$ slows down, meaning that new surfaces $S_i$ are smaller. Existing intra-thallus surfaces are subdivided into smaller areas by new hyphae, which corresponds to the beginning of the bulk densification phase of the network. This transition occurs approximately after 15 h of growth in standard condition (condition 0) and is delayed to 22 h in constrained environment (osmotic

**Table 4. Exponential fit of the temporal evolution of $S_i$.**

| Condition | $t_{min}$ [h] | $t_{max}$ [h] | $\alpha$ [$h^{-1}$] |
|:---:|:---:|:---:|:---:|
| (0) | 9.2 | 16.0 | $0.541 \pm 0.014$ |
| (1) | 13.2 | 25.9 | $0.280 \pm 0.005$ |
| (2) | 11.9 | 23.6 | $0.430 \pm 0.009$ |
| (3) | 10.4 | 21.1 | $0.318 \pm 0.006$ |
| (4) | 13.4 | 25.3 | $0.279 \pm 0.006$ |
| (5) | 7.6 | 14.0 | $0.610 \pm 0.016$ |

An exponential function $\beta \exp(\alpha t)$ was used to fit the evolution of $S_i$ on a range of values between $t_{min}$ and $t_{max}$ for each condition 0 to 5. Both $t_{min}$ and $t_{max}$ are dependent on $t_{0,i}$. $\alpha$ is the growth rate exponent with its uncertainties (one standard deviation). The $R^2$ values are in the range 0.97–0.99.

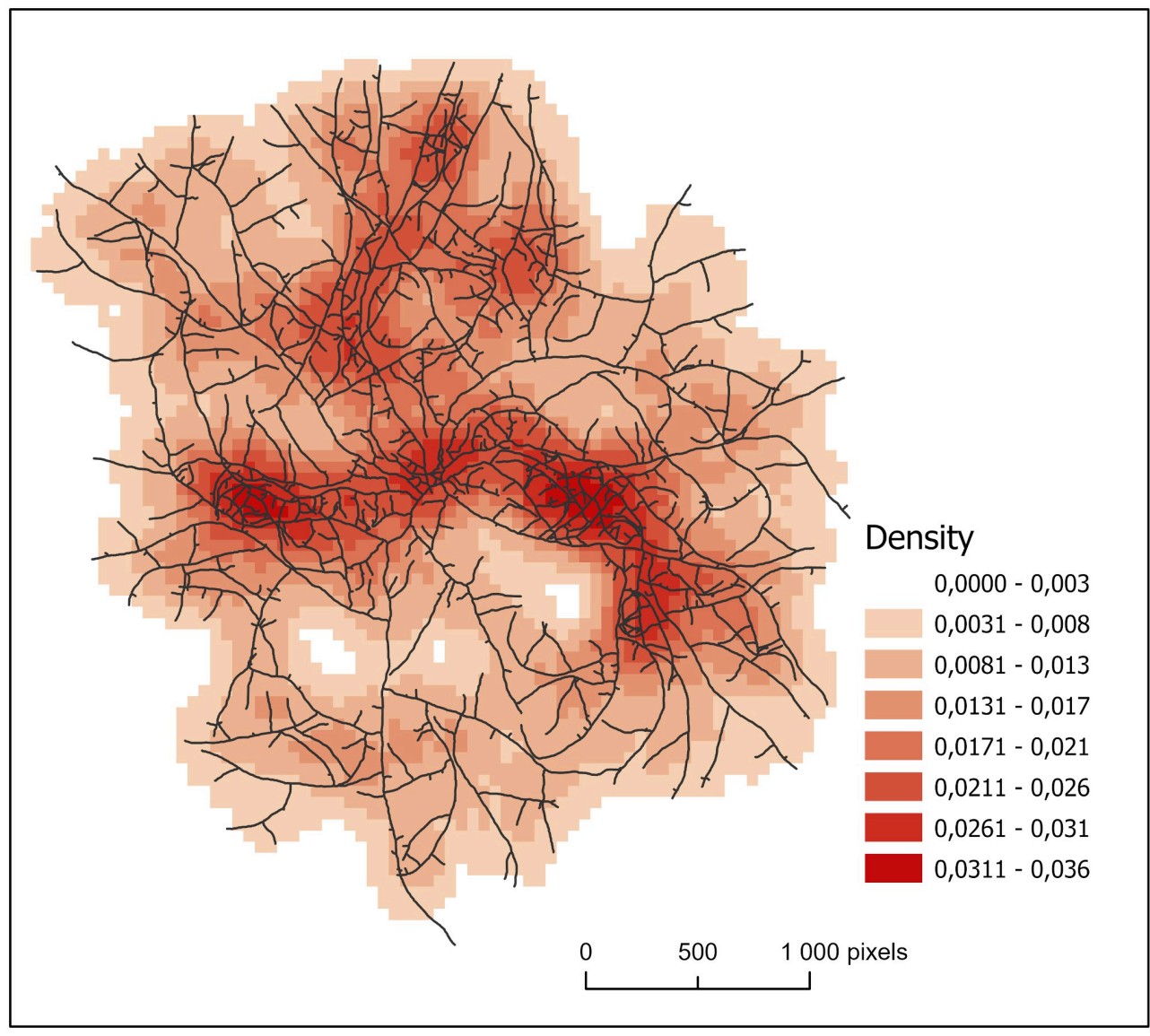

**Fig 7. Line densities extracted from a panorama after 15 h of growth in standard conditions (condition 0).** Densities have been discretized into eight classes. The hyphal network is shown in overprint. Cell size: 50 px; search radius: 200 px (see text).

stress, condition 2). It can be assumed that the bulk densification process is delayed in this constrained environment because creating new branches is costly in energy and material, and cannot be ensured immediately when the mycelium faces a stress. It should be noted that under the other conditions, the start of the bulk densification phase is not clearly detectable.

**Densities of line.** The study of densities of line provides a complementary approach to characterize the organization of the hyphal network. In Fig 7, we show the density values obtained on the fungal thallus in standard conditions (condition 0), after 15 h of growth, distributed into eight density ranges. The superimposed network makes it possible to clearly discern the match between cells with high line density values and regions where hyphae are numerous. We also notice the high-density areas (in red), described above as surfaces of

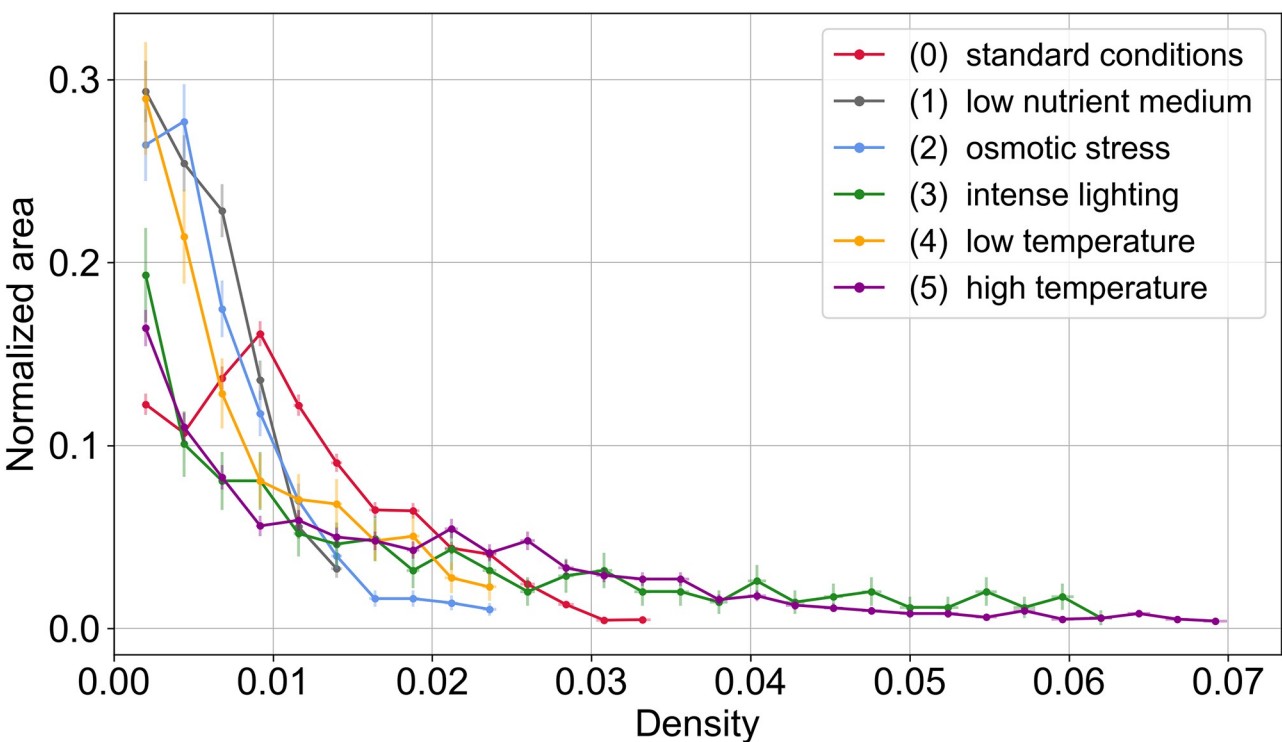

**Fig 8. Probability density function of the number of cells in the same line density range as a function of density, for mycelia after 15 h of growth.**
See Fig 7 and text for details. Bin size: 0.0024.

preferential densification, probably given by the orientation of the initial apexes (see Section Intra-thallus areas).

For each condition, at 15 h of growth, we built the density distribution (Fig 8) as the number of cells, normalized by the total surface, *i.e.* the envelope, as a function of their density value. The envelope is defined here as all non-zero pixel values of line density. All six curves have the same exponentially decreasing trend, indicating that cells of low density represent the majority of the surface in all cases. It can be assumed that these low-density areas correspond largely to the outer ring of the mycelium, where hyphae are still few in number. The proportion of areas of very low densities (less than 1%) is almost twice as high for conditions 1, 2 and 4 compared to conditions 3 and 5. Note that although the trends of the curves are approximately the same for the stressed conditions, the curve for condition 0 indicates that the proportion of cells with a low density (up to 1.5%) is higher than for the other conditions.

It is also interesting to compare the maximum density value reached for each condition at this growth time. Although their proportions are small, densities nearly twice as high as the control (0) are reached for the conditions in high light intensity (condition 3) and high temperature (condition 5) (about 6% and 7% respectively, against 3.2% for the control). Conversely, the conditions in low-nutrient environment (condition 1), in osmotic stress (condition 2) and at low temperature (condition 4) reach maximum densities lower than the control (condition 0).

## Discussion

### Macroscopic *versus* microscopic observations of the fungal thallus

In biology, the characterization of a growing fungal thallus has been limited for a long time to very classical macroscopic approaches, such as measuring the diameter of the thallus or monitoring the weight of the mycelium over several days [16]. Although informative, it is obvious that these descriptive approaches are of limited interest. Such measurements do not reflect the true complexity of the mycelium, and do not allow to follow the growth of different species, particularly in the early stages of growth, when the thallus is barely visible to the naked eye [9, 11]. Microscopic approaches have allowed for direct access to the structure of the fungal network itself, whose organization is fundamental for the survival of the fungus in its environment. The question that arises is how to quantify a growing fungal network in a robust way. Until around 2010, studies at the microscopic scale mainly reported manual readings on regions of the mycelium, or on an image at a given time (see for example, [6]). In recent years, various devices have been developed with the aim of extracting dynamic quantities characteristic of the fungal network from images of the mycelium. Studies are then carried out at the hypha scale (see for example [10, 14]), or at the mycelium scale in two or three dimensions [9, 11, 29]. In a previous work [17], we developed a reproducible experimental procedure to track the growth of the *P. anserina* fungal network in a controlled environment, though simple and reliable quantities, such as the number of apexes, the number of nodes and the total length of the mycelium. Such a device has enabled us to highlight the optimization of the angular branching process and the occurrence of different growth phases in the fungal network of *P. anserina* [18, 30].

### Various constraints applied *in vitro* on the fungal thallus

In this study, we undertook to better understand how a fungal network adapts to potentially unfavorable environmental changes. These can then cause stresses that will disrupt cellular homeostasis and could cause molecular damage to the cell [20]. We have chosen five stresses, simple enough to be controlled *in vitro*, with the same level of control as the standard culture conditions classically used for optimal growth and reproduction of *P. anserina* (condition 0), and biologically relevant, *i.e.* that can be encountered *in vivo*. One can easily imagine that in nature, a fungus must adapt to periods or areas where (i) nutrient resources are scarce or not readily available (condition 1), (ii) osmotic or light intensity conditions are locally non-optimal for growth (conditions 2 and 3, respectively), and (iii) temperatures are lower (condition 4) or higher (condition 5) than the optimal growth temperature. For each constraint, we monitored the development of the fungal thallus at two levels, one at the macroscopic scale over 7 days of growth, the other at the scale of the hyphae of the network, *i.e.* microscopic, over approximately 20 hours of growth. In the latter case, the architecture of the network was finely characterized by (i) the extraction of the growth quantities $A$, $N$ and $L$ and (ii) an original approach using geomatic tools, giving access to the dynamics of densification of the network, *via* the follow-up of intra-thallus surfaces and line densities.

### The limits of macroscopic observations and extraction of $A$, $N$ and $L$ quantities

In our study, we considered the condition 0 as a control, taking the optimal parameters for the growth of *P. anserina*, ensuring both a maximal growth and promoting the sexual reproduction on Petri dish, *i.e. in vitro* [16]. At the macroscopic level, all the constraints applied on the fungal thallus affect growth, which is directly reflected in the reduced diameter of the thallus

during the whole observation period (7 days), especially under intense light and high temperature conditions. Typical characteristics of a mycelium under stress [16], such as thallus pigmentation or sparser hyphae are also observed, depending on the stresses applied, with intense lighting causing the most visible effects on the thallus. At the microscopic level, the simplest approach was to follow the quantities $A$, $N$ and $L$ over time, as described previously [17] or as it has been led in other fungi [8, 9, 11]. Tracking these quantities over time clearly discriminated the standard condition from all other conditions but did not reveal significant differences between the tested constraints. In summary, we have shown here that the production of material (hyphae) at the thallus level is clearly affected by the different stresses, but that the monitoring of the quantities $A$, $N$ and $L$ over time is not sufficient to finely characterize the effect of each stress on the organization of the network, particularly at the level of the densification dynamics.

## The interest of geomatic tools in accessing the densification process

For this purpose, geomatic tools usually used by geographers have been adapted to our study: the mycelial network is then analyzed as a map; we are interested not in the material (hyphae produced), but in the empty surfaces between these hyphae. This approach, which was initiated in our previous work [17] has been further developed here. We have shown that the total surface area of the mycelium $S$ expands at different rates over time depending on the condition, and that the latent phase observed at the beginning of growth is always followed by a linear phase. The extension of the thallus in the standard condition is the fastest, and the thallus that has undergone light stress is the most affected. In addition, it appears that the growth under any of these stresses increases the duration of the lag phase, *i.e.* the period of time before the beginning of the linear phase (as described in Fig 6A). This can be explained by the fact that during this lag time, the fungus must adapt its metabolism to the locally perceived environment. For example, in the case of an osmotic stress where the water balance is disturbed, we know that the fungus resists by synthesis and accumulation of intracellular osmolytes such as glycerol or mannitol to adjust its internal solute potentials [19, 20]. This osmoregulation may take time to take place, thereby increasing the duration of the lag phase, compared to a situation without stress. Furthermore, we have shown that the number of intra-thallus surfaces $S_i$ increases exponentially over time for all conditions, with different growth rate exponents, the control and high temperature conditions being the ones with the highest exponents. The processes of both total surface extension and densification are differentiated according to the conditions. The thallus can cover a large surface and be sparse (condition 1), or conversely be small in total surface and very dense (condition 3). The conditions can be classified in an order according to the evolution of $S$ (coefficient $a$), but in a different order based on the evolution of $S_i$ (coefficient $\alpha$). Moreover, by combining the study of the total surface $S$ and the number of intra-thallus surfaces $S_i$, we have access, in some cases, to the beginning of the bulk densification phase, which typically occurs after 15 h of growth in standard condition (condition 0). This fundamental step which signs the massive start of densification is delayed in the case of osmotic stress and is not clearly highlighted in the case of other applied stresses, thus marking a less marked start of the process. The line densities, studied here at a fixed time (15 h of growth), show that intense lighting and high temperature induce mycelia twice as dense as the control with a very reduced occupied surface. This could be related to an increased frequency of connections and an earlier start of densification. On the contrary, on a medium poor in carbon source, the fungal thallus extends rather well but is very sparse. In the first case where mycelia are very dense, it is likely that the stress applied on the thallus creates an environment too hostile for exploratory growth and the hyphae then develop on a reduced operating area.

From a physiological point of view, previous studies revealed that the effects of light were observed in polysaccharide, fatty acid, nucleotide and nucleoside metabolisms and in the regulation of production of secondary metabolites [21]. Unsurprisingly, such effects severely affect thallus growth and organization, as seen in this work. Conversely, when nutrient resources are scarce, it is likely that the fungus increases its capacity to take up nutrients by expanding as much as possible while maintaining a sparse mycelium [19]. It should be noted here that the M0 medium does not contain any carbon source but it is known that *P. anserina* is able to partially degrade the cellophane used to maintain the fungus in 2D [16]. However, this carbon source is in greatly reduced quantity compared to the standard M2 medium. Finally, it is surprising to note that a temperature above (+8˚C) or below (−8˚C) the standard temperature does not cause the same effects on the fungal network, as it can be seen at all levels (macro scale, number of apexes, nodes, total length, intra-thallus surfaces, line densities). Such observations have already been made in [11] who showed that, at fixed relative humidity, the mycelial area and the number of tips of *C. puteana* and *R. solani* both displayed different dynamics for the four tested growth temperatures. This suggests that the physiological consequences on the fungal thallus are different depending on whether the temperature is too low or too high. As maps and metrics extracted from intrathallus areas as well as density maps revealed to be useful for monitoring thallus growth and how it densifies, additional GIS spatial analyses may be done to highlight these spatio-temporal dynamics, based for example on spatial autocorrelation (Global Moran's I). Performed on thallus segments or intrathallus areas, this method would allow us to assess their clustered, dispersed or random pattern [31, 32]. In the same way, hot spot / cold spot analyses performed on nodes, thallus segments of intrathallus-areas using the Getis-Ord Gi* statistic tool would be helpful for identifying statistically the spatial distribution of the fungus [32].

## From *in vitro* experiments to a complex *in vivo* environment

Overall, in this study, we have shown that, under different constraints, the production of material that constitutes the fungal network is globally slowed down compared to standard conditions and that the densities of the networks are more or less highly depending on the tested condition. Moreover, thanks to the micro and macroscopic approaches carried out in parallel, it appears that the characteristics specific to each network in the first hours of growth are maintained over time and will durably affect the development of the fungal thallus after several days of growth. However, we must keep in mind that the *in vitro* conditions applied here are rather far from a complex *in vivo* environment. Indeed, here, to simplify the analysis, and as a first intention, we have selected five constraints, applied separately and over the whole observation period. It could be interesting to extend our study to other abiotic stresses, which fungi are likely to encounter in their environment, such as variations in pH [20], or humidity [11], the presence of polluted water (toxic metals) or hydrophobic compounds [33]. Hyphae can also be subjected to mechanical constraints, such as obstacles in their pathways, or confinement in the pores of the growth substrate [34, 35]. Moreover, filamentous fungi evolve in habitats that are sometimes highly coveted by other organisms, and competition for substrate colonization between the different species then takes place. The mycelium can be attacked by organisms such as nematodes [36] or springtails [12]. This is particularly true for microbial communities that compete within a nutrient rich, yet physically patchy and ephemeral, resource, as *P. anserina* in the dung community [37]. Furthermore, in our experiments, the fungus grows in 2D on a homogeneous medium and is subjected to a single continuous stress. These highly controlled conditions are obviously very far from the complex natural environment of a fungus [3]. In nature, fungi must face stresses that can be punctual (in time and

space), repetitive and especially combinatorial [22, 38]. Fungi are indeed exposed to multiple stresses that act in combination rather than independently [11, 20, 39]. Moreover, it has been shown that the response of a fungus to a stress can also vary depending on whether it is a first confrontation or if the fungus has previously been able to adapt to a more or less similar stress. Also, the response to several combined stresses is not the simple sum of individual stress responses [33, 39]. It is therefore interesting to consider in the future to approach more realistic environmental conditions. Observing fungi in their natural habitats is often complex, as they usually grow in opaque environments such as soil or plant tissue. Most studies are based on laboratory cultures, on synthetic media and under controlled conditions. Some studies, however, have been carried out under conditions more faithful to the fungus real environment, notably in the case of large-scale mycelia such as wood decomposers. The latter can be observed in soil microcosms, composed of sieved, compressed and non-sterile soil, or directly in the field [13].

In conclusion, this study contributes to a better understanding how filamentous fungi adapt the growth and densification of their network to potentially adverse various environmental changes and thus face evolutionary pressure driving natural selection. Then, the plasticity of the hyphal network enables the mycelium to be reorganized in order to ensure the survival of the fungus in the face of combinatorial stresses. The architecture of the network could then be considered as partly a consequence of the growth environment.

## Supporting information

**S1 Table. Parameter values of the exponential fit for the three replicates in conditions 0 to 5.** The exponential fit is $X_i(t) = X_i^0\, 2^{(t+t_{0,i})/\tau_i}$, where $X$ stands respectively for $A$, $N$ or $L$ and $i$ for experiment $i = 1, 2$ or 3. $t$ is time and $t_{0,i}$ is the temporal offset. $\tau_i$ is the characteristic growth time. The uncertainty represents one standard deviation.
(PDF)

## Acknowledgments

We warmly thank Sylvie Cangemi and Aurélien Renault for their expert technical assistance.

## Author Contributions

**Conceptualization:** Éric Herbert, Florence Chapeland-Leclerc.

**Funding acquisition:** Éric Herbert, Florence Chapeland-Leclerc.

**Investigation:** Clara Ledoux, Cécilia Bobée, Éva Cabet, Frédéric Filaine, Sabrina Hachimi.

**Methodology:** Clara Ledoux, Éric Herbert, Florence Chapeland-Leclerc.

**Project administration:** Éric Herbert, Florence Chapeland-Leclerc.

**Supervision:** Éric Herbert, Florence Chapeland-Leclerc.

**Validation:** Pascal David, Christophe Lalanne, Gwenaël Ruprich-Robert.

**Writing – original draft:** Clara Ledoux, Cécilia Bobée, Éric Herbert, Florence Chapeland-Leclerc.

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
