## [Decision Letter · Decision Letter 0]

26 Jul 2023

PONE-D-23-03797Characterization of spatio-temporal dynamics of the constrained network of the filamentous fungus Podospora anserina using a geomatics-based approachPLOS ONE

Dear Dr. Chapeland-Leclerc,

Thank you for submitting your manuscript to PLOS ONE!  I want to start out by apologizing for the time it has taken to provide this initial review.  I was asked to step in after another editor had to pull out and it took some time to identify reviewers.

After careful consideration, we feel that your manuscript has merit but does not fully meet PLOS ONE’s publication criteria as it currently stands. Therefore, we invite you to submit a revised version of the manuscript that addresses the points raised during the review process.

Please consider the suggestions of the 2 reviewers while preparing your revised manuscript. I believe that most of the comments can be addressed by providing additional details about your methodology in the text.

We look forward to receiving your revised manuscript!

Kind regards,

Katherine A. Borkovich, Ph.D.

Academic Editor

PLOS ONE

“This work is supported by IdEx Université Paris Cité (ANR-18-IDEX-0001) and by ANR-21-CE40010-01. https://anr.fr”

“We warmly thank Sylvie Cangemi and Aur´elien Renault for their expert technical 597 assistance. Clara Ledoux is supported by a PhD scholarship from the Doctoral School 598 MTCI (ED 563). This work is supported by the IdEx Universit´e Paris Cit´e 599 (ANR-18-IDEX-0001) and by ANR-21-CE40010-01.”

“This work is supported by IdEx Université Paris Cité (ANR-18-IDEX-0001) and by ANR-21-CE40010-01. https://anr.fr”

Reviewers' comments:

Reviewer's Responses to Questions

**Comments to the Author**

1. Is the manuscript technically sound, and do the data support the conclusions?

Reviewer #1: Yes

Reviewer #2: Yes

2. Has the statistical analysis been performed appropriately and rigorously? 

Reviewer #1: Yes

Reviewer #2: Yes

3. Have the authors made all data underlying the findings in their manuscript fully available?

Reviewer #1: Yes

Reviewer #2: Yes

4. Is the manuscript presented in an intelligible fashion and written in standard English?

Reviewer #1: Yes

Reviewer #2: Yes

5. Review Comments to the Author

Reviewer #1: In this work the authors build upon their previous work using image analysis to characterize several quantitative elements of thallus development in Podospora anserina. The authors grow a wild-type strain in five different stress conditions and a control condition to analyze the impact of different stresses on thallus development. The authors find that using quantitative measures such as number of apexes, nodes and total length of mycelium can distinguish the stress conditions from the control but not stress conditions from each other. To better distinguish between the stress conditions the authors supplement their previous imaging analysis with geomatics-based analysis to characterize the effect of stresses on mycelial organization.

Overall, this work is well constructed, so I have no major issues:

Minor issues:

1. Some lines could be edited for clarity.

a. rows 67-68: […] combined to a fierce competition…

b. rows 255-256

c. The claim in line 258 is that the thallus in condition 1 was most dense, but then on line 261 the thallus in condition 4 is described as extremely dense.

d. row 436: […] small density…

e. row 536: […] little dense…

2. In the Results section: Apexes, nodes and length growth dynamics following the different symbols were difficult for the equation was difficult as several symbols are quite similar and there are many, perhaps a table could be added to make it easier to follow the text or the text could be modified to be clearer what each symbol stands for and where it was derived from (lines 289-311).

Reviewer #2: General comments:

The authors' multidisciplinary approach in studying the construction of the hyphal network of P. anserina is interesting, especially with the use of image processing and geomatics. The manuscript is well-written and aligns with the authors' previous studies. While I lack expertise in lab methods and this specific fungi ecology, I appreciate the innovative GIS-based pluridisciplinary methodological approach presented here that has the potential to be explored for other fungi. To improve the article's clarity, consider moving some parts from the results to the "Material and Methods" and “Discussion” sections, as well as linking the results more explicitly with other studies in the "Discussion" section.

Abstract:

• The abstract could benefit from providing more detailed information about the methods used. This would help readers better understand the study's scope right from the start.

Introduction:

• Consider rephrasing lines 86-88 “Then, five different constraints commonly used for the growth of P. anserina in vitro have been tested in this study and compared to optimal standard conditions”.

• Provide clearer explanations for "strong lighting" on line 92-93, such as specifying whether it refers to UV light or a particular wavelength or something else.

• Line 98: Uppercase correction for “Information”

• Avoid redundancy in lines 98-101 by summarizing the concept of GIS and its relevance to the study more concisely. Modification suggestion: “GIS are computer system tools used in the geomatics discipline, and involve a wide range of methods for collecting, managing and analyzing data gathered by various devices (including satellites, drones, meteorological stations, governmental census…).”

Materials and methods:

• Update the figure reference on line 180 to be "Fig 1" as it appears first in the text.

• Eliminate redundancy by removing the text in brackets on lines 183-184, as the abbreviation for "GIS" has already been defined in the introduction.

• Specify the geomatics software used and its version on lines 182-185 for better reproducibility.

• For better accessibility to a broader audience, consider adding the main functions used at each step of the geomatics approach, including software-specific functions like "function Intersect" used in ArcGIS.

Results:

• In Table 1, clarify "Standard conditions" in brackets or replace "Condition 1" with "Standard conditions (see [17])" to avoid confusion. Also, ensure the table is readable independently from the text and consistent with the information in the introduction (you talked about 5 conditions + standard conditions in the introduction while here you talked about 6 conditions including standard conditions).

• As in the introduction, provide technical specifications for terms such as "lighting" and "luminosity" with "neutral" and "intense" either in the table legend or directly in the table.

• Move several parts of the text (lines 239-247, 249-254, 272-274, 277-301, 334-345, 358-369, 421-425) to the "Material and methods" section. Consider moving Table 1 to “Material and Methods” section as well.

• Consider relocating some parts of the text (lines 254-257, 304-307, 323-331, 399-401) to the "Discussion" section since they are more focused on interpretation.

• Make sure to specify the condition "lighting" again on line 260, as you did for other conditions below, to enhance clarity.

• Consider adding scale bar texts ("10mm", "500 um", "50um") to Figures 2 and 3.

• Clarify the text on lines 282 (Fig 3A?) and 283 (Fig 3B?), referring to the relevant figures explicitly.

• Again, clearly state the number of conditions tested (one standard condition and five abiotic constraint conditions) on line 298 to avoid confusion.

• In Figure 8, write out the abbreviation "Pdf" fully the first time it appears.

Discussion:

• Consider adding subsection titles for more clarity.

• Remove "previously described in [17]" from line 467 to avoid redundancy.

• The discussion should elaborate more on how the study complements existing research on the topic and consider citing relevant studies to support your claims (e.g., at lines 479-481 and 539-541).

• Based on my understanding, P. anserina was selected for this research due to its ease of cultivation in the laboratory. It serves as a proof of concept, mainly focused on developing the geomatics approach to study hyphal network dynamics. The objective is to demonstrate how this method can be applied not only to P. anserina but also to various other fungi in both in vitro and potentially in vivo scenarios. I think the discussion section requires a better explanation of the implications of the study more explicitly (lines 482-484), particularly regarding its relevance to in vivo conditions and the significance of testing environmental constraints on this specific fungus found on dung.

• P. anserina naturally evolves in a highly competitive habitat where other fungi, microbes, and various biotic factors may influence the dynamics of its hyphal network. I believe that lines 565-582 necessitate further discussion and elaboration concerning the potential consequences of excluding biotic factors and the potential interplay between biotic and abiotic stressors. How the results might be impacted by the absence of biotic factors? And what disparities could arise between in vitro (with only one single continuous stress) and in vivo experiments (complex system)?

• Could further exploration be conducted on other geomatics approaches, such as pattern analysis using tools like the Moran I index, spatial autocorrelation, and cluster analysis?

Supporting information:

• Correct the spelling error "replicats" to "replicates" in S1 Table legend.

6. PLOS authors have the option to publish the peer review history of their article (what does this mean?). If published, this will include your full peer review and any attached files.

Reviewer #1: No

Reviewer #2: No

---

## [Author Response · Author response to Decision Letter 0]

7 Dec 2023

We thank the editor and reviewers for their kind and constructive comments, and feel that they have strengthened the manuscript. Responses to each comment are listed below and are shown in red.

“This work is supported by IdEx Université Paris Cité (ANR-18-IDEX-0001) and by ANR-21-CE40010-01. https://anr.fr”

“We warmly thank Sylvie Cangemi and Aur´elien Renault for their expert technical 597 assistance. Clara Ledoux is supported by a PhD scholarship from the Doctoral School 598 MTCI (ED 563). This work is supported by the IdEx Universit´e Paris Cit´e 599 (ANR-18-IDEX-0001) and by ANR-21-CE40010-01.”

“This work is supported by IdEx Université Paris Cité (ANR-18-IDEX-0001) and by ANR-21-CE40010-01. https://anr.fr”

For points 2 and 3: In the cover letter we mention that “Clara Ledoux is supported by a PhD scholarship from the Doctoral School MTCI (ED 563). This work is supported by the IdEx Université Paris Cité (ANR-18-IDEX-0001) and by ANR-21-CE40010-01. The funders had no role in study design, data collection and analysis, decision to publish, or preparation of the manuscript." As requested, we have removed funding-related text from the Acknowledgments Section of the manuscript.

Reviewer #1: 

In this work the authors build upon their previous work using image analysis to characterize several quantitative elements of thallus development in Podospora anserina. The authors grow a wild-type strain in five different stress conditions and a control condition to analyze the impact of different stresses on thallus development. The authors find that using quantitative measures such as number of apexes, nodes and total length of mycelium can distinguish the stress conditions from the control but not stress conditions from each other. To better distinguish between the stress conditions the authors supplement their previous imaging analysis with geomatics-based analysis to characterize the effect of stresses on mycelial organization.

Overall, this work is well constructed, so I have no major issues:

Minor issues: 

1. Some lines could be edited for clarity. 

The reviewer is right. To clarify the points raised, we have modified the manuscript as follows:

a. rows 67-68: […] combined to a fierce competition…

We propose the following change, as we feel the two sentences are clear enough as written: “P. anserina evolves in a highly competitive habitat where several dozens of species are present and feed on partially degraded plant material. So, growth of P. anserina in this confined microcosm appears to be highly challenged by resource-limited and patchy dung environment (Silar et al., 2020).”

b. rows 255-256

After careful review, we finally decided to delete the two sentences mentioned, which add nothing to the macroscopic description of the fungal thalli studied. 

c. The claim in line 258 is that the thallus in condition 1 was most dense, but then on line 261 the thallus in condition 4 is described as extremely dense. 

The sentence in brackets on line 261 was unclear and has been reworded as follows: “most extensive thallus, fairly dense, regular, not pigmented mycelium”. Thallus in condition 3 is then described as extremely dense, compared to condition 0. 

d. row 436: […] small density…

Unlike point c above which dealt with observations made on a macroscopic scale, point d concerns microscopic observations from which line densities have been extracted on a thallus scale. In the manuscript, the term “small” has been replaced by “low”, which we believe is more correct.

e. row 536: […] little dense…

In the manuscript, “but is very little dense” has been replaced by “but is very sparse”, which is make more sense.

2. In the Results section: Apexes, nodes and length growth dynamics following the different symbols were difficult for the equation was difficult as several symbols are quite similar and there are many, perhaps a table could be added to make it easier to follow the text or the text could be modified to be clearer what each symbol stands for and where it was derived from (lines 289-311).

We thank the reviewer who pointed out the lack of clarity of this paragraph. The text has been rewritten, and some notations were removed. According to the reviewer 2, this paragraph has been moved to materials and methods section.

Reviewer #2: 

General comments: The authors' multidisciplinary approach in studying the construction of the hyphal network of P. anserina is interesting, especially with the use of image processing and geomatics. The manuscript is well-written and aligns with the authors' previous studies. While I lack expertise in lab methods and this specific fungi ecology, I appreciate the innovative GIS-based pluridisciplinary methodological approach presented here that has the potential to be explored for other fungi. To improve the article's clarity, consider moving some parts from the results to the "Material and Methods" and “Discussion” sections, as well as linking the results more explicitly with other studies in the "Discussion" section. 

Abstract:

• The abstract could benefit from providing more detailed information about the methods used. This would help readers better understand the study's scope right from the start.

The reviewer is right. To be more precise about the original methods used in this work, we modified the abstract as followed: 

“….Thanks to an original geomatics-based approach based on the use of automatized Geographic Information System (GIS) tools, we were able to produce maps and metrics characterizing the growth dynamics of the networks and then to highlight some very different dynamics of network densification according to the applied stresses…”. 

Introduction:

• Consider rephrasing lines 86-88 “Then, five different constraints commonly used for the growth of P. anserina in vitro have been tested in this study and compared to optimal standard conditions”. 

In fact, the terms "commonly used" referred to standard conditions and not to the various stresses tested in vitro. We propose to delete these terms, resulting in the following sentence: “Then, several constraints have been tested in this study and compared to optimal standard conditions for the growth of P. anserina in vitro:”

• Provide clearer explanations for "strong lighting" on line 92-93, such as specifying whether it refers to UV light or a particular wavelength or something else.

“strong lighting” was replaced by “intense lighting of visible light”. More details are given in Materials and Methods. 

• Line 98: Uppercase correction for “Information”

Done.

• Avoid redundancy in lines 98-101 by summarizing the concept of GIS and its relevance to the study more concisely. Modification suggestion: “GIS are computer system tools used in the geomatics discipline, and involve a wide range of methods for collecting, managing and analyzing data gathered by various devices (including satellites, drones, meteorological stations, governmental census…).”

Done.

Materials and methods:

• Update the figure reference on line 180 to be "Fig 1" as it appears first in the text. 

After re-reading, we have decided to delete the reference to this figure, which makes no contribution here.

• Eliminate redundancy by removing the text in brackets on lines 183-184, as the abbreviation for "GIS" has already been defined in the introduction.

Done.

• Specify the geomatics software used and its version on lines 182-185 for better reproducibility. 

• For better accessibility to a broader audience, consider adding the main functions used at each step of the geomatics approach, including software-specific functions like "function Intersect" used in ArcGIS. 

According to the referee, the following sentences were added in the relevant paragraph: “The software used was ArcGIS Pro (version 3.0.3) and automation of treatments was carried out with the model builder interface. In this section, the terms below in square brackets correspond to the main GIS tools used”. 

Results:

• In Table 1, clarify "Standard conditions" in brackets or replace "Condition 1" with "Standard conditions (see [17])" to avoid confusion. Also, ensure the table is readable independently from the text and consistent with the information in the introduction (you talked about 5 conditions + standard conditions in the introduction while here you talked about 6 conditions including standard conditions).

For more clarity, in the introduction, we replaced "five different constraints" by "several constraints". Then, throughout the manuscript, we checked that we are always talking about 6 conditions, including condition 0 which is the reference standard condition, referring to reference (17) when necessary, as suggested by the reviewer. We indicated that “standard conditions” indeed refer to “Condition 0” (in Table 1 and Figure 2) and we added the relevant reference in the caption of Table 1 and Figure 2. 

• As in the introduction, provide technical specifications for terms such as "lighting" and "luminosity" with "neutral" and "intense" either in the table legend or directly in the table.

We added the following sentence “The light intensities for neutral and intense lighting

was measured to 67 lux and 66.103 lux, respectively” in the caption of Table 1.

• Move several parts of the text (lines 239-247, 249-254, 272-274, 277-301, 334-345, 358-369, 421-425) to the "Material and methods" section. Consider moving Table 1 to “Material and Methods” section as well. 

After careful rereading of the results and the reviewers' proposals, it appears that some sentences or paragraphs are indeed repetitions of the "materials and methods" section and have been deleted from the "results" section and, in some cases, moved to the "materials and methods" section. In this way, most of the reviewers' requests have been taken into account. However, for the sake of clarity, we prefer to keep Table 1 and certain sentences or paragraphs in the "results" section. We hope that this new organization will suit the reviewer.

239-247: we've left the paragraph and table setting out all the conditions to this section.

249-254: this paragraph was partially deleted.

272-274: this paragraph was deleted.

277-301 and 334-345: almost all have been moved to “materials and methods” section.

358-369: we've left the paragraph to this section.

421-425: this paragraph was deleted.

• Consider relocating some parts of the text (lines 254-257, 304-307, 323-331, 399-401) to the "Discussion" section since they are more focused on interpretation.

254-257: this point did not seem clear or relevant to the other reviewer. After re-reading, we finally decided to delete the two aforementioned sentences, which add nothing to the macroscopic description of the fungal thalli studied. 

304-307: this short paragraph has been left in the "results" section, as this rather technical point is not included in the discussion. 

323-331 and 399-401: these paragraphs were removed from the results, but they remain included in the discussion anyway.

• Make sure to specify the condition "lighting" again on line 260, as you did for other conditions below, to enhance clarity.

Done.

• Consider adding scale bar texts ("10mm", "500 um", "50um") to Figures 2 and 3.

Done.

• Clarify the text on lines 282 (Fig 3A?) and 283 (Fig 3B?), referring to the relevant figures explicitly.

Done.

• Again, clearly state the number of conditions tested (one standard condition and five abiotic constraint conditions) on line 298 to avoid confusion.

Done.

• In Figure 8, write out the abbreviation "Pdf" fully the first time it appears.

Done

Discussion:

We would like to thank the reviewer for his constructive comments on the discussion, which has now been extensively revised to improve its clarity and relevance.

• Consider adding subsection titles for more clarity.

The discussion has been reorganized and is now divided into five sections.

• Remove "previously described in [17]" from line 467 to avoid redundancy.

Done.

• The discussion should elaborate more on how the study complements existing research on the topic and consider citing relevant studies to support your claims (e.g., at lines 479-481 and 539-541).

We have now added references and explanations concerning some experimental observation systems and image analyses reported to date of a fungal thallus at microscopic scale, and the contribution of our experimental system (see section “Macroscopic versus microscopic observations of the fungal thallus), emphasizing the interest of the geometric tools deployed in this work (see section “The interest of geomatic tools in accessing the densification process”).

• Based on my understanding, P. anserina was selected for this research due to its ease of cultivation in the laboratory. It serves as a proof of concept, mainly focused on developing the geomatics approach to study hyphal network dynamics. The objective is to demonstrate how this method can be applied not only to P. anserina but also to various other fungi in both in vitro and potentially in vivo scenarios. I think the discussion section requires a better explanation of the implications of the study more explicitly (lines 482-484), particularly regarding its relevance to in vivo conditions and the significance of testing environmental constraints on this specific fungus found on dung.

• P. anserina naturally evolves in a highly competitive habitat where other fungi, microbes, and various biotic factors may influence the dynamics of its hyphal network. I believe that lines 565-582 necessitate further discussion and elaboration concerning the potential consequences of excluding biotic factors and the potential interplay between biotic and abiotic stressors. How the results might be impacted by the absence of biotic factors? And what disparities could arise between in vitro (with only one single continuous stress) and in vivo experiments (complex system)?

We have taken care to answer these questions, particularly in section " From in vitro experiments to a complex in vivo environment " which has been enriched with new references and explanations, in relation to the reviewer's questions. In particular, we have detailed the interest of studying a copropilous fungus like Podospora anserina which evolves in an extremely competitive habitat and we discuss the interest and the limits of the experiments carried out here in vitro, in a controlled but extremely simplified environment, in relation to the environmental conditions and the complexity of the microbiome in which P. ansrina naturally evolves. Future studies, including observation systems closer to in vivo conditions as well as other fungal species, should be considered in the future.

• Could further exploration be conducted on other geomatics approaches, such as pattern analysis using tools like the Moran I index, spatial autocorrelation, and cluster analysis? 

We agree with the reviewer and we add a specific paragraph about this point at the end of the section “The interest of geomatic tools in accessing the densification process”: 

Supporting information:

• Correct the spelling error "replicats" to "replicates" in S1 Table legend.

Done.

---

## [Editor Report · Decision Letter 1]

15 Jan 2024

Characterization of spatio-temporal dynamics of the constrained network of the filamentous fungus Podospora anserina using a geomatics-based approach

PONE-D-23-03797R1

Dear Dr. Chapeland-Leclerc,

We’re pleased to inform you that your manuscript has been judged scientifically suitable for publication and will be formally accepted for publication once it meets all outstanding technical requirements.

Kind regards,

Katherine A. Borkovich, Ph.D.

Academic Editor

PLOS ONE
---

## [Editor Report · Acceptance letter]

28 Jan 2024

PONE-D-23-03797R1 

PLOS ONE

Dear Dr. Chapeland-Leclerc, 

I'm pleased to inform you that your manuscript has been deemed suitable for publication in PLOS ONE. Congratulations! Your manuscript is now being handed over to our production team.

Kind regards, 

on behalf of

Dr. Katherine A. Borkovich 

Academic Editor

PLOS ONE